

# Identification of m6A-associated autophagy genes in non-alcoholic fatty liver

Ziqing Huang, Linfei Luo, Zhengqiang Wu, Zhihua Xiao and Zhili Wen

The Second Affiliated Hospital of Nanchang University, Nanchang, China

## ABSTRACT

**Background**. Studies had shown that autophagy was closely related to nonalcoholic fat liver disease (NAFLD), while N6-methyladenosine (m6A) was involved in the regulation of autophagy. However, the mechanism of m6A related autophagy in NAFLD was unclear.

**Methods**. The NAFLD related datasets were gained *via* the Gene Expression Omnibus (GEO) database, and we also extracted 232 autophagy-related genes (ARGs) and 37 m6A. First, differentially expressed ARGs (DE-ARGs) and differentially expressed m6A (DE-m6A) were screened out by differential expression analysis. DE-ARGs associated with m6A were sifted out by Pearson correlation analysis, and the m6A-ARGs relationship pairs were acquired. Then, autophagic genes in m6A-ARGs pairs were analyzed for machine learning algorithms to obtain feature genes. Further, we validated the relationship between feature genes and NAFLD through quantitative real-time polymerase chain reaction (qRT-PCR), Western blot (WB). Finally, the immuno-infiltration analysis was implement, and we also constructed the TF-mRNA and drug-gene networks.

**Results**. There were 19 DE-ARGs and four DE-m6A between NAFLD and normal samples. The three m6A genes and five AGRs formed the m6A-ARGs relationship pairs. Afterwards, genes obtained from machine learning algorithms were intersected to yield three feature genes (TBK1, RAB1A, and GOPC), which showed significant positive correlation with astrocytes, macrophages, smooth muscle, and showed significant negative correlation with epithelial cells, and endothelial cells. Besides, qRT-PCR and WB indicate that TBK1, RAB1A and GOPC significantly upregulated in NAFLD. Ultimately, we found that the TF-mRNA network included FOXP1-GOPC, ATF1-RAB1A and other relationship pairs, and eight therapeutic agents such as R-406 and adavosertib were predicted based on the TBK1.

**Conclusion**. The study investigated the potential molecular mechanisms of m6A related autophagy feature genes (TBK1, RAB1A, and GOPC) in NAFLD through bioinformatic analyses and animal model validation. However, it is critical to note that these findings, although consequential, demonstrate correlations rather than cause-and-effect relationships. As such, more research is required to fully elucidate the underlying mechanisms and validate the clinical relevance of these feature genes.

Corresponding author
Zhili Wen, wenzhili1@163.com

## INTRODUCTION

Non-alcoholic fatty liver disease (NAFLD) is a major cause of chronic liver disease, with a worldwide prevalence of approximately 25%. It is characterized by the accumulation of excessive triglycerides and other lipids in the hepatocytes (*Powell, Wong & Rinella, 2021*). NAFLD is a progressive disease that can progress from simple steatosis to steatohepatitis and eventually lead to cirrhosis or hepatocellular carcinoma (*Kanwal et al., 2018*). Owing to the current rapid changes in lifestyles, NAFLD is a public health issue and poses a great clinical and economic burden on the patient. Moreover, it has become the most prevalent liver disorder in China. China has been reported to have the fastest-growing prevalence of NAFLD in the world, with 314.58 million NAFLD individuals projected by 2030. It is also estimated that the population of NAFLD will increase from 80 million to 110 million in the USA by 2030 (*Estes et al., 2018*; *Zhou et al., 2020a*). However, the harmfulness and severity of NAFLD have not been paid enough attention. Currently, there are no standardised proposals by any country to address NAFLD at national and global levels (*Lazarus et al., 2022*).

NAFLD is a series of diseases involving excessive liver fat deposition, often accompanied by various metabolic disorders, there is a lack of specific therapeutic drugs (*Rong et al., 2022*). Non-pharmacological therapies that involve changes in diet and lifestyle are generally used, or indirect regulation and improvement of specific pathogenic factors, key links in the onset of the disease, and related metabolic disorders. For example, diosgenin (improving mitochondrial function), pomegranate (anti-inflammatory), curcumin (antioxidant) (*Chen et al., 2023*; *Zamanian et al., 2023*; *Beheshti Namdar et al., 2023*). The latest research shows that the interaction and regulation between serum metabolites and intestinal flora may help naringenin's therapeutic effect on NASH (*Cao et al., 2023*), and related studies targeting the intestinal-liver axis and regulating intestinal flora are constantly emerging (*Zhai et al., 2023*; *Wang et al., 2023*). On the other hand, another research evaluated disease-related circRNA and competitive endogenous RNA networks as potential biomarkers for NAFLD using functional gain and loss methods, reflecting the important role of targeted biomarker research in the progression of NAFLD (*Zeng et al., 2024*). Thus, we urgently need to elucidate the potential mechanisms of NAFLD.

Autophagy could maintains intracellular environmental homeostasis in response to external stimuli (*Hazari et al., 2020*), which is especially relevant to liver metabolism (*Byrnes et al., 2022*). Studies have shown that the dysregulation of autophagy is one of the important causes of NAFLD. Thus, the inhibition of autophagy can lead to an increase in the lipid droplet content of hepatocytes (*Cheng et al., 2019*). On the other hand, N6-methyladenosine (m6A) is the most pervasive internal modification of mRNA, which includes writers, erasers and readers (*He et al., 2019*). m6A affects the expression of target genes to maintain cellular functions and physiological processes by regulating mRNA processing, translation, degradation and splicing (*Liu & Gregory, 2019*; *Liu et al., 2017*). Currently, researchers have reported that the m6A modification plays an important role in the development of cancers and metabolic diseases (*Yang et al., 2019*; *Li et al., 2020*), and also in fatty liver disease (*Luo et al., 2019*). Importantly, it was reported that the autophagy

gene is closely associated with the m6A gene and fat mass and obesity (FTO), that the m6A demethylase can influence autophagy by decreasing the expression of ATG5 and ATG7 genes (*Wang et al., 2020*). Moreover, m6A writer methyltransferase like 3 (METTL3) and the m6A readers YTH N6-methyladenosine RNA binding protein 1 (YTHDF1) are negatively regulated autophagy pathway in NAFLD (*Peng et al., 2022*). Other studies also report that METTL3 regulates the m6A modification of lipid metabolism in hepatic cells and autophagy in cardiomyocytes (*Xie et al., 2019*). Considering the strong association between the m6A regulator and autophagy, we hypothesised that m6A-related autophagy genes may have important significance in NAFLD.

In this study, we aimed to screen the potential key m6A-related autophagy genes in NAFLD through comprehensive researches, and used bioinformatics tools to explore their potential function on NAFLD, including corresponding relationship with clinical features, immune features and transcription factor (TF) regulatory network. Further, the NAFLD animal model was built for the expression validation of key genes *in silico*. We make the case that the study could provide new biomarkers and a theoretical basis for the clinical diagnosis and treatment of NAFLD.

## MATERIALS & METHODS

### Data retrieved

The training set GSE66676 and validation set GSE130970 were downloaded *via* the Gene Expression Omnibus (GEO) (https://www.ncbi.nlm.nih.gov/gds) database, in which the GSE66676 contained 33 NAFLD samples and 34 normal samples (samples type: liver wedge biopsy) (*Xanthakos et al., 2015*), and GSE130970 validation set contained 72 NAFLD samples and three normal samples (samples type: liver biopsy) (*Hoang et al., 2019*). Additionally, 232 autophagy-related genes (ARGs) were extracted from the human autophagy database (HADb) (*Chen et al., 2022*) (http://www.autophagy.lu/), and 37 m6A were obtained from the literature (*Li et al., 2022*).

### Differentially expressed ARGs and differentially expressed m6A genes were obtained using differential analysis

Based on the NAFLD and normal samples in the GSE66676 dataset, differentially expressed ARGs (DE-ARGs) and differentially expressed m6A genes (DE-m6A) were sifted out by limma package (v3.46.0) (*Wang et al., 2021*) setting $P < 0.05$. Meanwhile, volcano maps were plotted for the obtained results. Heat maps were drawn using the pheatmap package (v1.0.12) (*Zhang et al., 2021a*) to visualize the expression patterns of DE-ARGs and DE-m6A.

### Construction of an m6A-ARGs co-expression network

Pearson correlation analysis was applied to analyse the relationships between DE-ARGs and DE-m6A using the cutoff Pearson correlation coefficient (PCC) $>0.7$ and $P < 0.05$, in which the gene selected were defined as the m6A-related autophagy genes (m6A-ARGs). The co-expression network of m6A-ARGs was drawn using the Cytoscape software. Then, WoLF-PSORT (https://wolfpsort.hgc.jp/) was used to predict the protein subcellular

localization (PSL) encoded by the m6A-ARGs. Finally, gene ontology (GO) and Kyoto Encyclopaedia of Genes And Genomes (KEGG) enrichment analysis of the m6A-ARGs were performed *via* the enrichGO and enrichKEGG functions in clusterprofiler R package (v3.18.1) for functionality annotations (*Yu et al., 2012*).

### Screening and validation of feature genes

Support vector machine-recursive feature elimination (SVM-RFE) and random forest (RF) are increasingly used to predict disease-associated feature genes. We narrowed down the feature genes using SVM-RFE and RF on the m6A-ARGs using Caret R package (v6.0-92). The resultant genes obtained were intersected to obtain the feature genes. Further, we used the rank sum test setting the threshold $P < 0.05$ to evaluate the expression levels of the feature genes between NAFLD and normal samples in the GSE66676 and GSE130970 datasets, respectively. Finally, gene set enrichment analysis (GSEA) of the feature genes was performed setting the GO and KEGG as reference gene sets, where the correlation coefficient of the average expression of all genes in all samples and feature genes was calculated as the ranking standard, and then the gseGO (parameter settings: OrgDb = org.Hs.eg.db, ont = 'ALL', $p$valueCutoff = 0.05, $p$AdjustMethod = 'BH') and gseKEGG (parameter settings: organism = 'hsa', $p$valueCutoff = 0.05, pAdjustMethod = 'BH') functions in the clusterProfiler R package (v3.18.1) were conducted to perform GSEA analysis on the sorted genes. The entries that meet the conditions of |NES | >1, $P < 0.05$ and FDR <0.25 were considered to have significant meaning to be selected.

### Correlation analysis of feature genes and clinicopathological characters

Based on the GSE66676 dataset, we analysed the correlation between feature genes and clinicopathological characters. The expression levels of the feature genes in different clinical states (borderline nonalcoholic steatohepatitis (NASH), definite NASH, NAFLD, not NASH, no NAFLD) were analysed using the Wilcoxon test. A box plot was drawn to visualize the results.

### The mice model of NAFLD was constructed

Male C57/12 mice (12 weeks old) which were purchased from Guangdong Vital River Laboratory Animal Technology Co., Ltd. (Guangdong, China), were housed in animal care facilities with controlled temperature (21~25 °C) and humidity (40~70%), and light-dark cycles were 12 h. Twenty mice were randomly divided into normal group ($n = 10$) and NAFLD group ($n = 10$). The NAFLD group were fed 60% high fat diet (HFD), the other group with normal diet ,and the weight of these mice were monitored weekly, and their food intake were recorded accordingly. Mice do not suffer during the feeding process, after feeding for 12 weeks, they were euthanized painlessly (cervical dislocation under anesthesia 2% isoflurane), their liver tissue was taken out for examination. To ensure the successful establishment of a high-fat model, we conducted a pre-experiment before the formal experiment. The animal experiment was approved by the ethical review committee of The Second Affiliated Hospital of Nanchang University, the approval number: NCULAE-20221031008. All of the animal procedures in this study were in accordance with

the Laboratory animal-Guideline for ethical review of animal welfare promulgated by Standardization Administration of China and with the ARRIVE guidelines.

## Hematoxylin-eosin staining

The liver sample was dealed according to the hematoxylin-eosin (HE) staining General processing procedure, Liver section morphological differences between normal and NAFLD groups in C57 mice were observed by HE staining.

## Quantitative reverse transcriptase-PCR

qPCR verified the expression of RAB1A, TBK1, GOPC genes in two groups of male C57 mice. Using the Trizol method to extract the total RNA of liver tissue according to the manufacturer's instructions, the concentration of each RNA sample was detected. Then, the total RNA was converted to complementary deoxyribonucleic acid (cDNA) according to the the reverse transcription reagent kit. The quantitative PCR kit TB Green was used for cDNA for real-time PCR detection with the procedure: denaturation temperature 96 °C, annealing temperature 57 °C, extension temperature 72 °C for detection. Primers of qRT-PCR were listed in Table 1.

## Western blotting detected the liver RAB1A, TBK1, GOPC protein content of C57 mice in the two group

The same weight of liver sample each group was taken and lysed with RIPA buffer to extracted the total proteins, which were quantified by using a microplate reader. Adding 1 µl of 5XSDS-PAGE protein loading buffer per 4 µL of protein sample before polyacrylamide gel electrophoresis, the subsequent routine western blot experimental steps are carried out, after the membranes were washed and successively incubated with the primary antibodies and the secondary antibody. The membrane was scanned on the Odyssey Li-COR CLx infrared laser scanner to obtain the content of RAB1A, TBK1, GOPC of rat liver.

## Immune infiltration analysis

Body immunity plays an important role in the occurrence and development of diseases. Immune infiltration analysis was performed on the GSE66676 dataset, whereas xCell was used to observe the percentage distribution of immune cell types in each sample. Then, immune cells with differential distribution between NAFLD and normal samples were screened using the Wilcoxon test, and they were taken Spearman correlation analysis with feature genes by setting $P < 0.05$ and |cor | >0.3. A scatter plot was drawn to visualize the results.

## Construction of TF-mRNA network

TFs are a group of protein molecules that can be combined with specific sequences of genes to ensure that feature genes are expressed at a specific time and space. Differential TF-feature gene (TF-mRNA) relationship pairs were extracted from the hTFtarget database (https://ngdc.cncb.ac.cn/databasecommons/database/id/6946), and we also constructed a TF-mRNA regulatory relationship network according to the above relationship pairs.

**Table 1 Primers of q-PCR.**

|  | Forward primer | Reverse primer |
|---|---|---|
| **Tbk1 (mouse)** | TGTTCTAGAGGAGCCGTCCA | GGTGCACTATGCCGTTCTCT |
| Rab1a **(mouse)** | ATCGTTTCCCGTGGTTGGTT | ACACTGGTTGTGCTGTGTGA |
| GOPC **(mouse)** | CACTCTGTGGAGGATCTGGAAA | CTCGCCCCATAAACTTCAGC |

## Drug forecast analysis

To predict potential therapeutic drugs associated with feature genes, we uploaded the feature genes into the Drug-Gene Interaction database (DGIdb) (http://www.dgidb.org). Default parameters were used to analyse the interaction between drugs and feature genes. The results were visualized using Cytoscape.

## Statistic analysis

Statistical analysis was carried out through R software. Differences between different groups were compared *via* the Wilcoxon test. $P < 0.05$ was considered as significant difference.

# RESULTS

## Identification of DE-ARGs and DE-m6A

The flowchart of data analysis is shown in Fig. 1. A total of 19 DE-ARGs was observed between NAFLD samples and normal samples, of which 15 were upregulated and four were downregulated DE-ARGs (Fig. 2A, Table 2). We also sifted out four DE-m6As (two up-regulated DE-m6As and two down-regulated DE-m6As) by differential expression analysis (Fig. 2B, Table 3).

## m6A-ARGs co-expression network

Using the Pearson correlation analysis of DE-ARGs and DE-m6A (Fig. 3A), a total of three m6A genes, namely PCIF1, HNRNPA2B1 and SRSF10, and five ARGs, namely ATG4D, ATG5, TBK1, GOPC and RAB1A were obtained as the m6A-ARGs with PCC >0.7 and $P < 0.05$. The co-expression network visualized the interactions of PCIF1-ATG5, SRSF10-GOPC, HNRNPA2B1-ATG4D and other relationship pairs (Fig. 3B). Afterwards, the detected protein localization-distribution results in WoLF-PSORT analysis was displayed in Fig. 3C, in which the proteins encoded by TBK1 and ATG5 were presented in the cytoplasm, while those encoded by RAB1A and ATG4D were found in the extracellular matrix. Additionally, GOPC-encoded proteins were located in the nucleus. The enrichment analysis revealed that all genes in the m6A-ARGs co-expression network was involved in 10 GO pathways, such as macro-autophagy and selective autophagy (Fig. 3D). Autophagy can affect the development of NAFLD by reducing the degradation of intracellular lipid droplet closure and lysosomal fusion. Likewise, the results of KEGG analysis indicated that various m6A-ARGs were associated with the pathway of autophagy-animal, and the top10 enriched KEGG terms were showed in Fig. 3E.

## Identification of feature genes

We respectively gained four and three genes *via* SVM-RFE and RF algorithms (Figs. 4A–4B), and they were intersected to yield three feature genes (TBK1, RAB1A and GOPC)
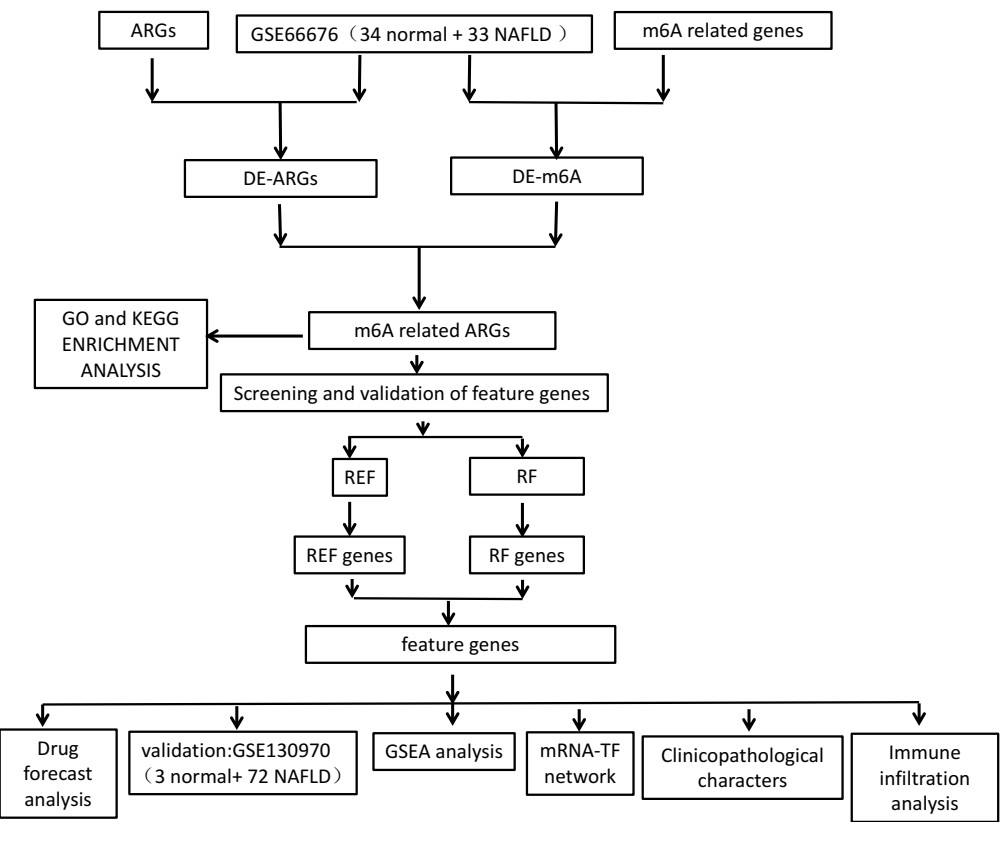

**Figure 1 Flowchart of overall study design.**

(Fig. 4C). According to the literature, TBK1 inhibitors alleviate the pathological response of NAFLD, suggesting that TBK1 is associated with NAFLD (*Oral et al., 2017*; *Huh et al., 2020*). Ras-related protein RAB1A is a member of the cellular G-protein Ras superfamily, which plays a role in protein transport and membrane remodeling. GOPC encodes a Golgi protein that had a PDZ structural domain, and Golgi proteins have been reported to play a role in intracellular protein transport and degradation. Nevertheless, the expression trends of three feature genes were the same in the GSE66676 (left) and GSE130970 (right) datasets, and they were significantly increased in NAFLD compared with normal samples (Fig. 4D).

## The expression of feature genes are higher in rats with NAFLD (*n* = 10) than normal group (*n* = 10)

HE was used to verify the successful construct an *in vivo* model of NAFLD (Fig. 5A), the mRNA expression of TBK1, RABA1, GOPC were all significantly increased in the NAFLD group compared with the normal group (Fig. 5B). Meanwhile, the protein expression of TBK1, RABA1, GOPC were also increased in the NAFLD group than the normal group (Figs. 5C–5D), the result showed that the three feature genes may play an important role in NAFLD.

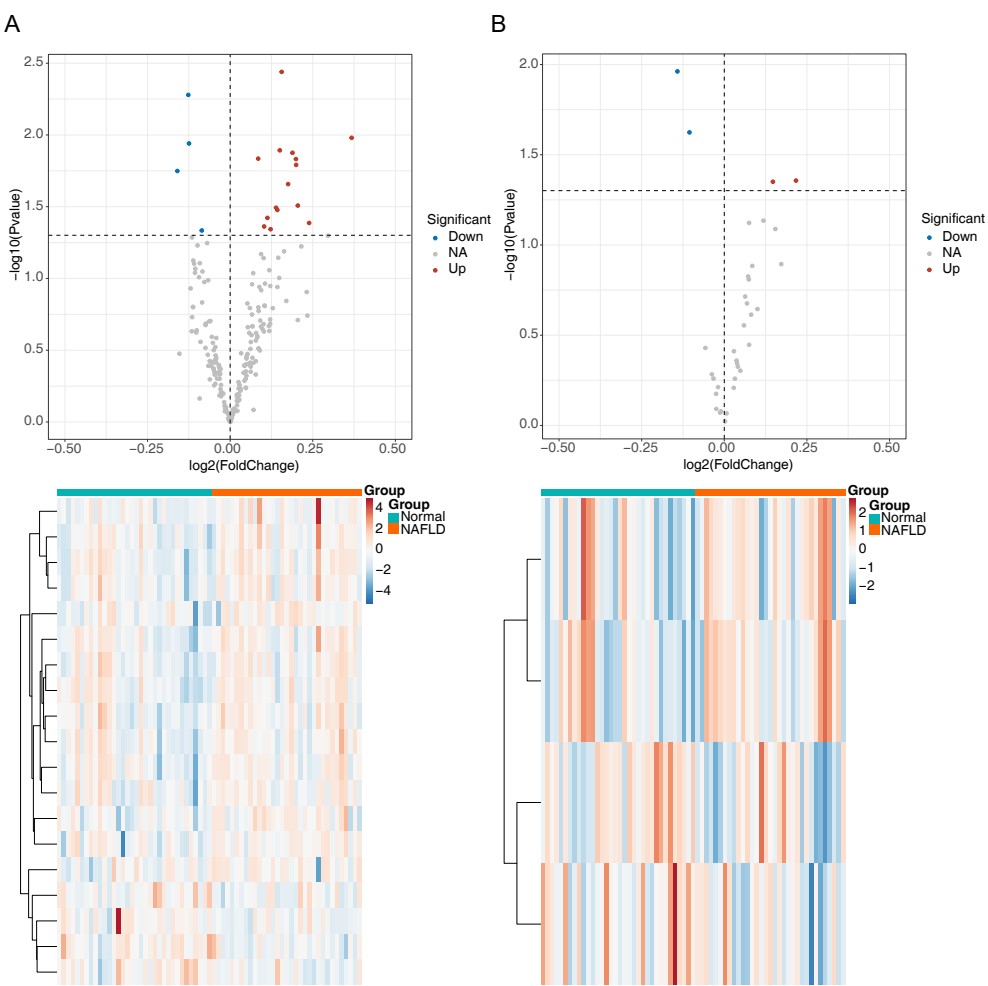

**Figure 2  Identification of differentially expressed autophagy related genes (DE-ARGs) and differentially expressed m6A genes (DE-m6A).** (A) Volcano plot (top) and heat map (bottom) of DE-ARGs between NAFLD and normal samples in GSE66676, red represents up-regulation, blue represents downregulation. (B) Volcano plot (top) and heat map (bottom) of DE-M6A genes in GSE66676. For A–B, $P <$ 0.05 was set as significant differences. The rows in the heat map represent the expression patterns of each gene in samples from different sources (green: Normal; orange: NAFLD), and the columns represent the expression patterns of different genes in each sample (blue: downregulation; red: upregulation), where the cluster tree on the left represents the genes with similar expression patterns were clustered.

## Enrichment and clinical correlation analyses of feature genes

TBK1 was enriched in 1,861 GO pathways, such as the catalytic step 2 spliceosome, and it was enriched to 150 KEGG pathways, including fatty acid degradation, protein export and so on (Fig. 6A). RAB1A was enriched in 1,408 GO pathways, including antigen processing and presentation of peptide antigen *via* MHC class I, and 108 KEGG pathways, including the proteasome pathway (Fig. 6B). A total of 2,068 GO pathways were enriched by GOPC, including the ribonucleoprotein complex biogenesis, and 149 KEGG pathways were obtained, including nucleocytoplasmic transport (Fig. 6C). Furthermore, the box

**Table 2  Lists for the differentially expressed autophagy related genes (DE-ARGs) between NAFLD and normal samples in GSE66676.**

|  | logFC | AveExpr | *t* | *P*.value | adj. *P*.val | B |
|---|---|---|---|---|---|---|
| CDKN1B | 0.155991255 | 7.791996413 | 3.009818825 | 0.00363408 | 0.381531286 | −2.054356556 |
| ATG4D | −0.126722986 | 6.893206907 | −2.881078481 | 0.005258718 | 0.381531286 | −2.309982934 |
| HSPA5 | 0.367701476 | 10.80340176 | 2.63104235 | 0.010464713 | 0.381531286 | −2.782569957 |
| ITGB4 | −0.12442223 | 5.509958325 | −2.596851071 | 0.011461408 | 0.381531286 | −2.844637478 |
| GOPC | 0.150314626 | 7.564719096 | 2.55511884 | 0.012794154 | 0.381531286 | −2.919533697 |
| EIF2AK3 | 0.188627977 | 6.914901924 | 2.540104874 | 0.013306904 | 0.381531286 | −2.946245686 |
| IKBKB | 0.084903034 | 7.375056448 | 2.503880459 | 0.01462123 | 0.381531286 | −3.010181302 |
| RHEB | 0.199168958 | 7.335165 | 2.501335584 | 0.01471782 | 0.381531286 | −3.014645578 |
| HSP90AB1 | 0.199743343 | 10.09404822 | 2.46475225 | 0.016171234 | 0.381531286 | −3.078420713 |
| PPP1R15A | −0.159745345 | 6.108658464 | −2.426477422 | 0.017828565 | 0.381531286 | −3.144337995 |
| TBK1 | 0.175433344 | 7.21801386 | 2.342607025 | 0.02200197 | 0.42803833 | −3.285855566 |
| RAB1A | 0.204874846 | 9.768151338 | 2.201047573 | 0.031037081 | 0.509247311 | −3.515394688 |
| WIPI1 | 0.138979281 | 7.753411242 | 2.186279158 | 0.032145563 | 0.509247311 | −3.538653978 |
| MAPK9 | 0.142781973 | 7.815001148 | 2.171171147 | 0.033315245 | 0.509247311 | −3.562311787 |
| HDAC6 | 0.112872637 | 8.079616328 | 2.116973775 | 0.037822661 | 0.5215983 | −3.646038191 |
| DNAJB9 | 0.239011167 | 7.756817388 | 2.081609315 | 0.041041743 | 0.5215983 | −3.699700039 |
| BECN1 | 0.102938592 | 7.101400105 | 2.056883887 | 0.043430886 | 0.5215983 | −3.736758985 |
| ATG5 | 0.122487529 | 7.607437523 | 2.037567537 | 0.045379987 | 0.5215983 | −3.765446156 |
| IFNG | −0.085846797 | 3.768942053 | −2.028600253 | 0.046310129 | 0.5215983 | −3.778684503 |

**Table 3  Lists for the differentially expressed m6A (DE-m6A) genes between NAFLD and normal samples in GSE66676.**

|  | logFC | AveExpr | *t* | *P*.value | adj. *P*.val | B |
|---|---|---|---|---|---|---|
| CDKN1B | 0.155991255 | 7.791996413 | 3.009818825 | 0.00363408 | 0.381531286 | −2.054356556 |
| ATG4D | −0.126722986 | 6.893206907 | −2.881078481 | 0.005258718 | 0.381531286 | −2.309982934 |
| HSPA5 | 0.367701476 | 10.80340176 | 2.63104235 | 0.010464713 | 0.381531286 | −2.782569957 |
| ITGB4 | −0.12442223 | 5.509958325 | −2.596851071 | 0.011461408 | 0.381531286 | −2.844637478 |
| GOPC | 0.150314626 | 7.564719096 | 2.55511884 | 0.012794154 | 0.381531286 | −2.919533697 |
| EIF2AK3 | 0.188627977 | 6.914901924 | 2.540104874 | 0.013306904 | 0.381531286 | −2.946245686 |
| IKBKB | 0.084903034 | 7.375056448 | 2.503880459 | 0.01462123 | 0.381531286 | −3.010181302 |
| RHEB | 0.199168958 | 7.335165 | 2.501335584 | 0.01471782 | 0.381531286 | −3.014645578 |
| HSP90AB1 | 0.199743343 | 10.09404822 | 2.46475225 | 0.016171234 | 0.381531286 | −3.078420713 |
| PPP1R15A | −0.159745345 | 6.108658464 | −2.426477422 | 0.017828565 | 0.381531286 | −3.144337995 |
| TBK1 | 0.175433344 | 7.21801386 | 2.342607025 | 0.02200197 | 0.42803833 | −3.285855566 |
| RAB1A | 0.204874846 | 9.768151338 | 2.201047573 | 0.031037081 | 0.509247311 | −3.515394688 |
| WIPI1 | 0.138979281 | 7.753411242 | 2.186279158 | 0.032145563 | 0.509247311 | −3.538653978 |
| MAPK9 | 0.142781973 | 7.815001148 | 2.171171147 | 0.033315245 | 0.509247311 | −3.562311787 |
| HDAC6 | 0.112872637 | 8.079616328 | 2.116973775 | 0.037822661 | 0.5215983 | −3.646038191 |
| DNAJB9 | 0.239011167 | 7.756817388 | 2.081609315 | 0.041041743 | 0.5215983 | −3.699700039 |
| BECN1 | 0.102938592 | 7.101400105 | 2.056883887 | 0.043430886 | 0.5215983 | −3.736758985 |
| ATG5 | 0.122487529 | 7.607437523 | 2.037567537 | 0.045379987 | 0.5215983 | −3.765446156 |
| IFNG | −0.085846797 | 3.768942053 | −2.028600253 | 0.046310129 | 0.5215983 | −3.778684503 |

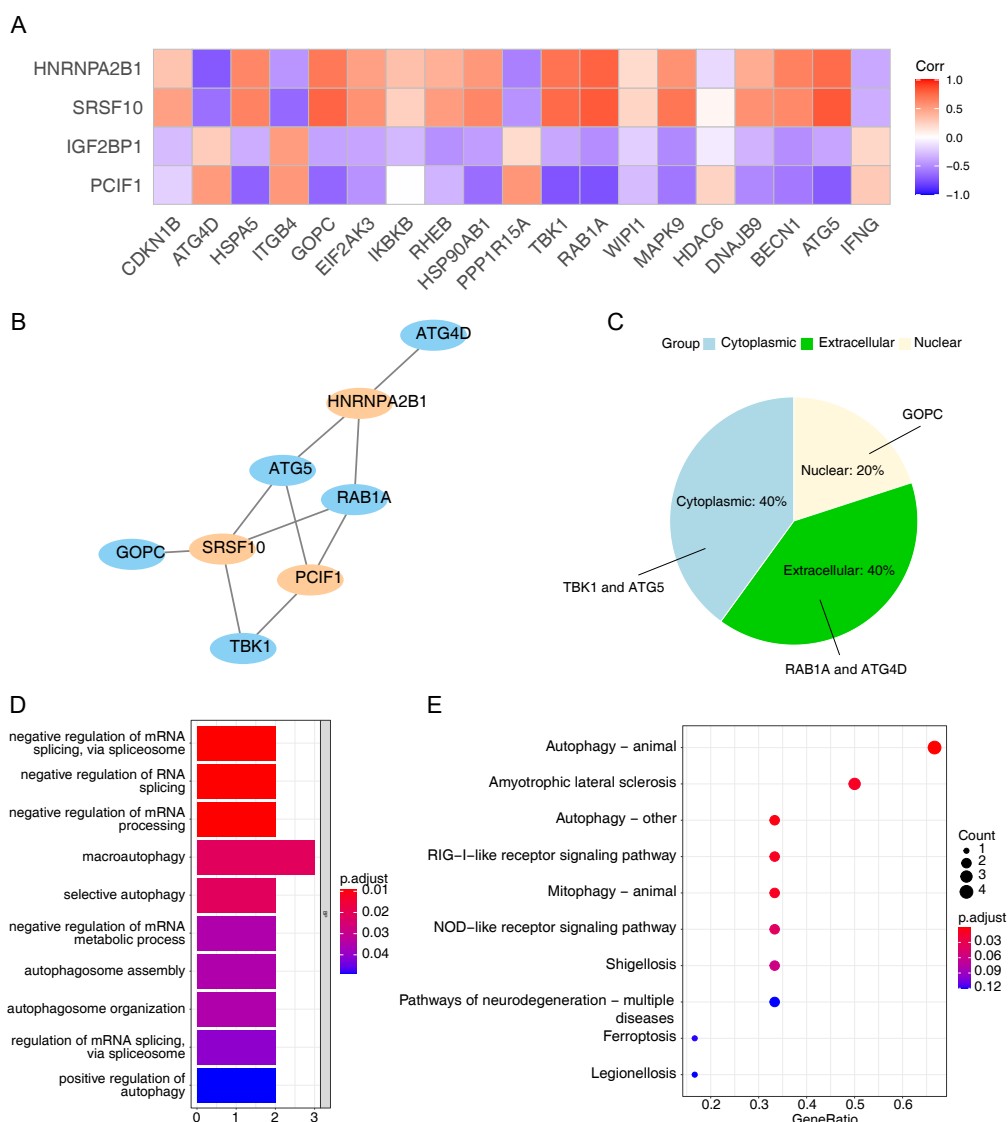

**Figure 3  Collection of m6A-related autophagy genes (m6A-ARGs).** (A) Pearson correlation heatmap of DE-ARGs and DE-m6A to screen m6A-ARGs using the cutoff pearson correlation coefficient (PCC) > 0.7 and $P < 0.05$. The abscissa is the DE-ARGs, the ordinate is DE-m6A. (B) Visualization of the co-expression network among m6A-ARGs *via* Cytoscape. Blue represents ARG and orange represents m6A gene. (C) Pie chart of WoLF-PSORT results for the predicted protein subcellular localization (PSL) which was encoded by m6A-ARGs. (D–E) Gene Ontology (GO) and Kyoto Encyclopaedia of Genes and Genomes (KEGG) enrichment analysis of all genes in the m6A-ARGs co-expression network.

plot revealed that the expressions of TBK1, RAB1A and GOPC in borderline NASH, define NASH, NAFLD, not NASH and no NAFLD were significantly different (Fig. 6D).

## Immune micro environment analysis

The percentage distribution of eight immune cells, such as astrocytes, chondrocytes, and epithelial cells, were all dramatically different between NAFLD and normal samples

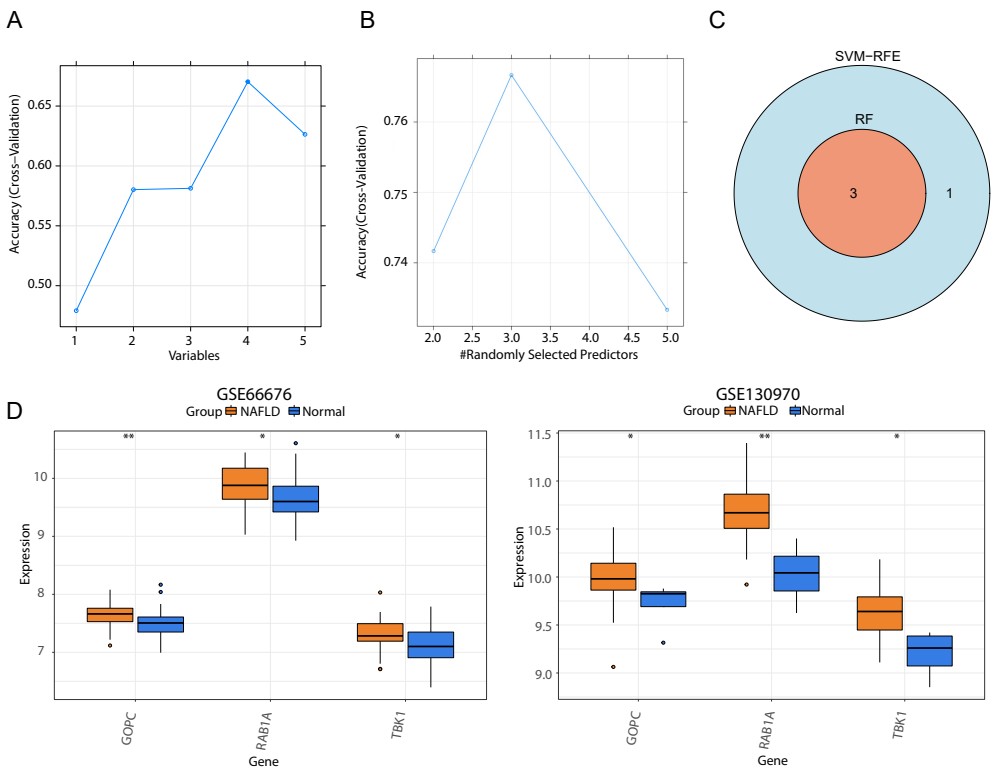

**Figure 4  Screening of three feature genes and expression verification in two public datasets.** (A) Four candidate genes were selected in the support vector machine-recursive feature elimination (SVM-RFE) model. The abscissa indicates the number of feature genes in RFE analysis, the ordinate represents the accuracy of the model. Blue line refers to the tendency of accuracy with the number of feature genes. (B) Three candidate genes were selected by the random forest (RF) model. (C) Venn diagram for three feature genes shared by two models. (D) Box plots of the expression levels of three feature genes between normal and disease samples in GSE66676 (left) and GSE130970 (right). **$p < 0.01$, *$p < 0.05$.

(Fig. 7A). Correlation analysis revealed that RAB1A, TBK1 and GOPC were significantly positively correlated with astrocytes, macrophages and smooth muscle but negatively correlated with epithelial cells and endothelial cells (Figs. 7B–7G).

## TF-mRNA regulatory relationship network

The GOPC, RAB1A and TBK1 was related to 37, 33 and 54 TF-mRNA pairs, respectively, and the TF-mRNA network included FOXP1-GOPC, ATF1-RAB1A, AR-TBK1 and other relationship pairs (Fig. 8).

## Drug sensitivity analysis

For the predicted therapeutic agents from the DGIdb database, data on GOPC and RAB1A was absence. Based on TBK1, we finally predicted eight therapeutic agents, namely R-406, adavosertib, chembl-1997335, PF-00562271, CYC-116, TAE-684, cenisertib and entrectinib (Fig. 9).
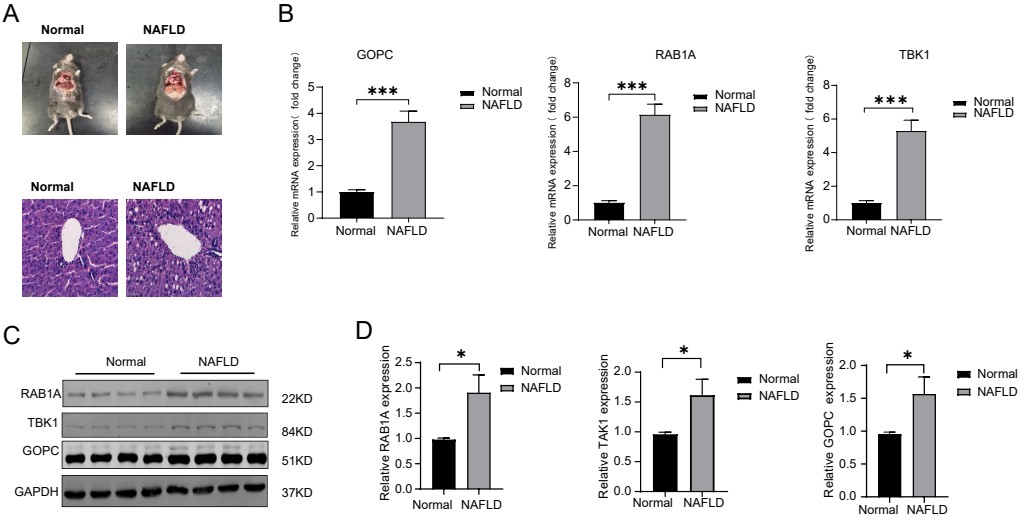

**Figure 5** **The mRNA and protein expression of TBK1, RAB1A and GOPC in animal model.** (A) Hematoxylin-eosin (HE) staining image as the pathological evidence for treatment of NAFLD. (B) Quantitative reverse transcriptase-PCR (qRT-PCR) showed the mRNA expression of the three feature genes were significantly higher in NAFLD rats ($n = 10$) than in normal rats ($n = 10$). (C–D) Western blotting (WB) showed that the three feature genes was significantly higher in NAFLD rats ($n = 5$) than in normal rats ($n = 5$). * $P < 0.05$, *** $P < 0.001$.

## DISCUSSION

M6A RNA methylation regulators play a role in preventing age-related and diet-induced development of NAFLD by improving inflammatory and metabolic phenotypes (*Peng et al., 2022*; *Qin et al., 2021*). However, knowledge of the biological role of m6A-related autophagy genes in the development of NAFLD is lacking. Recently research has shown autophagy is associated with the development of NAFLD, four autophagy-related lncRNAs were be found may participate in the occurrence of NAFLD (*Cao et al., 2022*). In this study, we explored differentially expressed m6A genes and autophagy genes in NAFLD and constructed an m6A-autophagy genes co-expression network using machine learning algorithms (SVM-RFE and RF) to filter out signature genes (*Chen & Ishwaran, 2012*; *Sanz et al., 2018*). Three feature genes, namely TBK1, RAB1A and GOPC were selected as the biomarkers of NAFLD.

TANK Binding Kinase 1 (TBK1), a protein-coding gene, is a critical kinase that modulates inflammation and autophagy. It was reported that the abnormal expression of TBK1 is related to obesity, diabetes, even and NAFLD (*He et al., 2020*; *Xu et al., 2018*). That is, as a member of the non-canonical IKK family, TBK1 is considered a by-product of activating the NF-$\kappa$B signalling pathway in the liver (*Zhao et al., 2018*). Blocking the IKK $\varepsilon$ and TBK1 pathways can significantly reduce the inflammatory factors produced by pro-inflammatory cells such as TNF-$\alpha$ and MCP-1, thereby improving insulin sensitivity and reducing liver steatosis (*Reilly et al., 2013*). TBK1 can reduce the thermogenesis and catabolism of the mitochondria by inhibiting the activity of AMPK, resulting in energy storage and ultimately

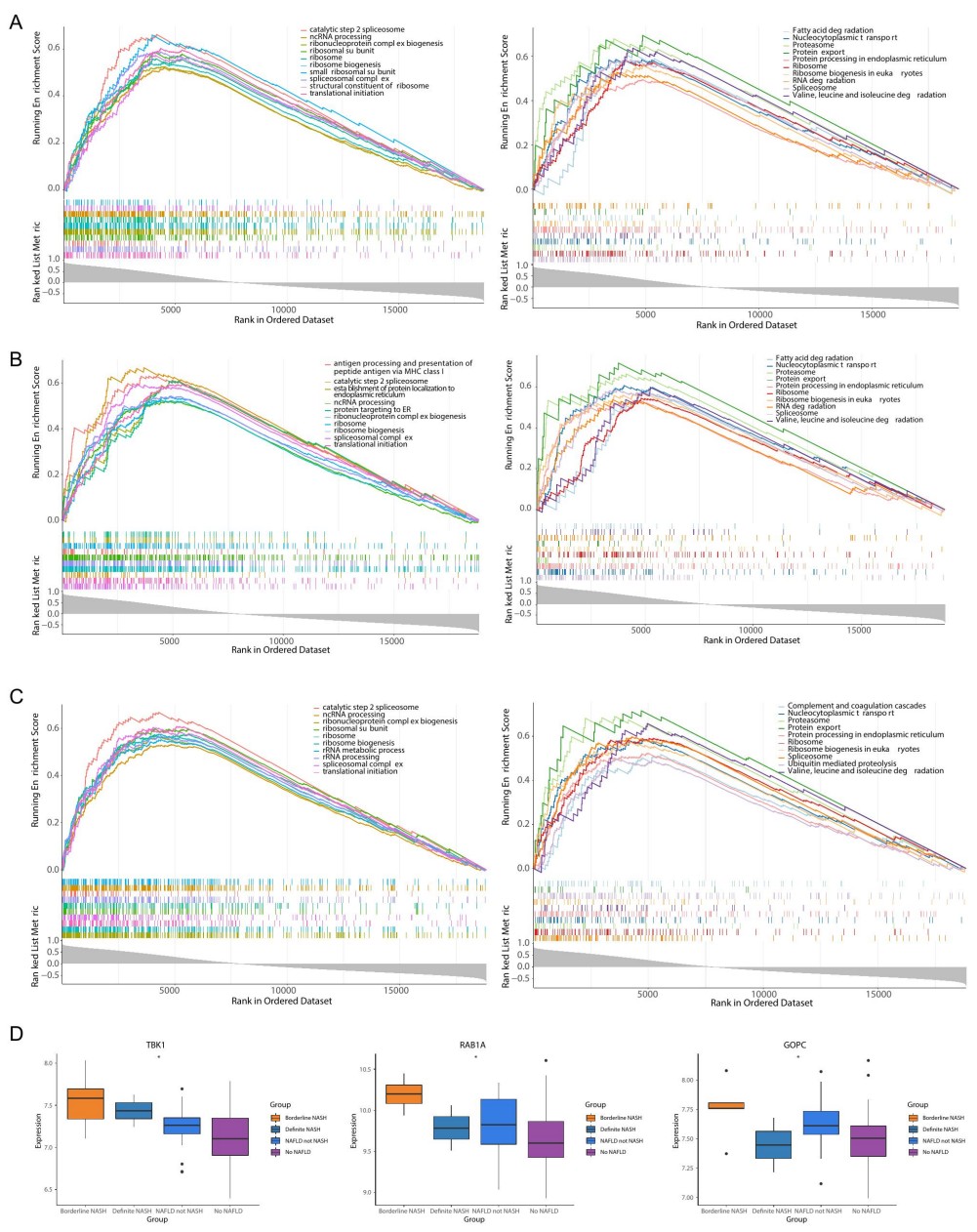

**Figure 6 Functionality and correlations with clinicopathological characters of three feature genes in the GSE66676 dataset.** (A–C) Gene set enrichment analysis (GSEA) of three feature genes (left: GO; right: KEGG). (A) *TBK1*. (B) *RAB1A*. (C) *GOPC*. Each figure shows partial enrichment entries (top 10). The figure contains three parts. The top of the figure refer to the enrichment score (ES) of each gene. A particularly obvious peak on the left is the ES value on the phenotype of the gene set. Each line in the middle of the figure represents a gene in the gene set and its ranking position in the gene list. The bottom of the figure shows a matrix of gene-phenotype correlations. (D) Box plots of the expression of *TBK1, RAB1A, GOPC* in different clinical states (borderline nonalcoholic steatohepatitis (NASH), definite NASH, NAFLD, not NASH, and no NAFLD. $^{***}p < 0.001$, $^{**}p < 0.01$, $^{*}p < 0.05$).

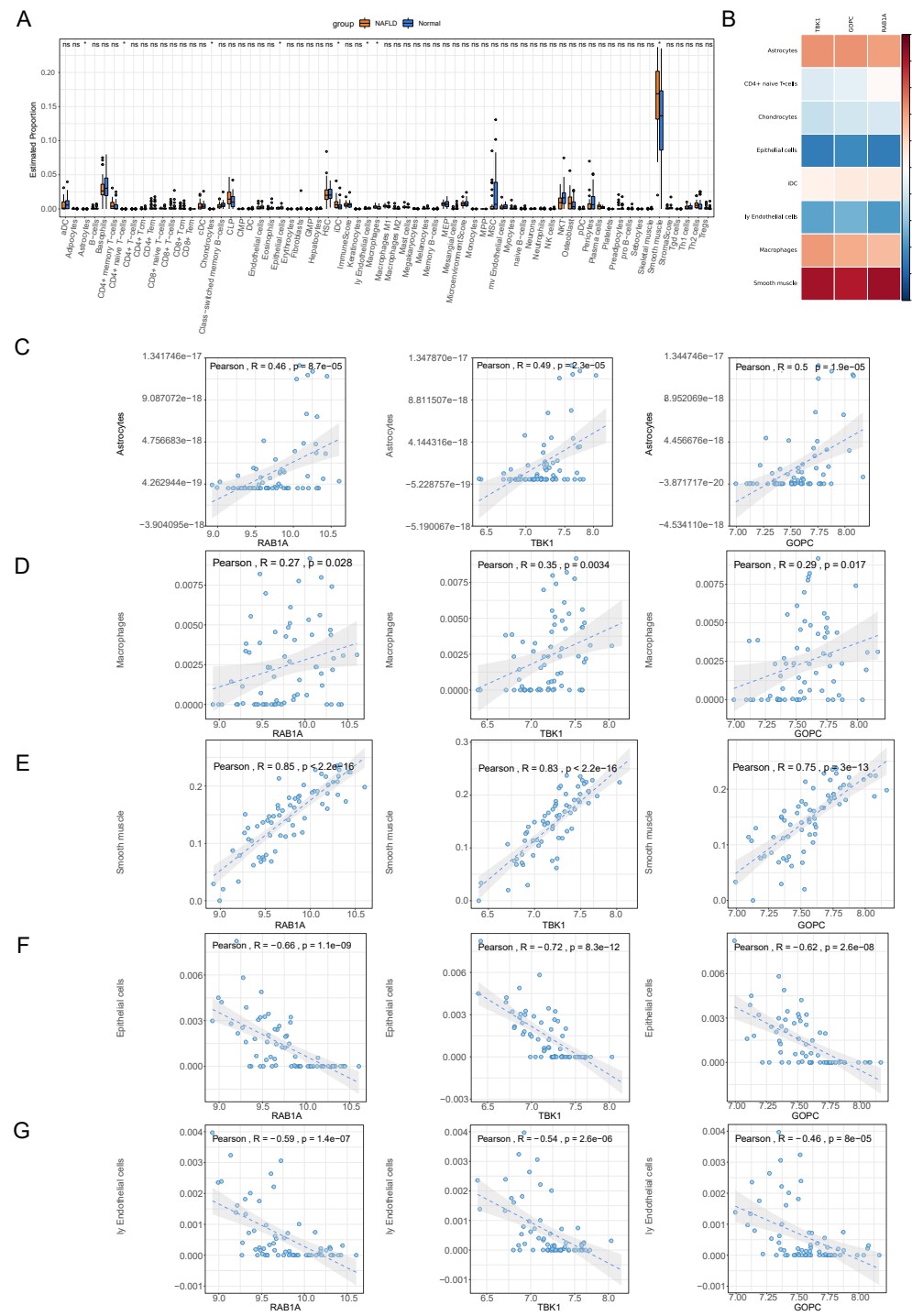

**Figure 7  Immune correlation analysis.** (A) Box plot for the proportions of immune cells subtypes in NAFLD and normal samples from the GSE66676 dataset (xCell algorithm). (B) Correlation heat map of eight differential expressed immune cells with feature genes. (C–G) Scatter plot for correlations of five significant immune cells with feature genes. (C) Astrocytes. (D) Macrophages. (E) Smooth muscle. (F) Epithelial cells. (G) Endothelial cells.

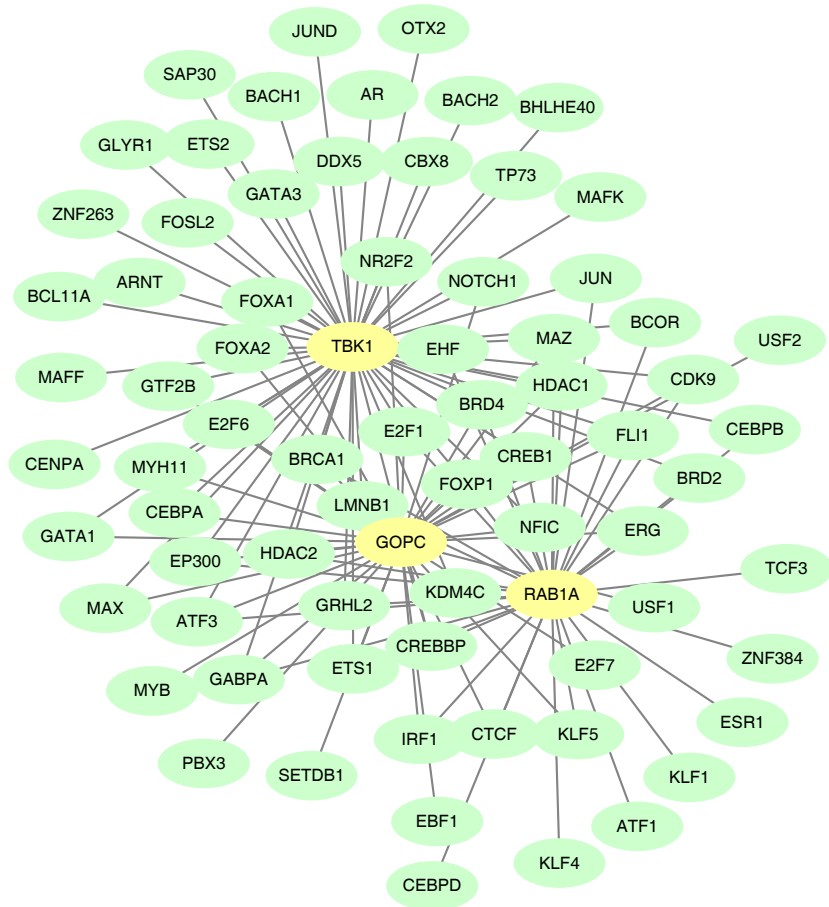

**Figure 8 Construction of the transcription factor (TF)-mRNA regulatory network.** Green is TF and yellow is feature gene.

leading to obesity (*Zhao et al., 2018*). It has been reported that TBK1 is increased in palmitic acid (PA)-treated liver cells (especially in liver Kupffer cells), suggesting its potential role in the progression of NAFLD to NASH. Corresponding to it, TBK1 inhibitors can alleviate PA-mediated lipid accumulation, inflammatory reaction and adipocyte apoptosis in hepatocytes (*Zhou et al., 2020b*), indicating the vital indicative significance of TBK1 in the progression of NAFLD.

RAB1A is notable for its role in vesicular trafficking and is generally considered to be a housekeeping protein that regulates cell membrane dynamics (*Yang et al., 2016*). The abnormal expression of RAB1A has been associated with many human diseases, such as glucose homeostasis, Parkinson's disease and various cancers (*Zhang et al., 2021b*; *Coune et al., 2011*; *Huang et al., 2021*). Autophagy generally provides a protective function by limiting tumour necrosis and inflammation, thereby reducing the damage of tumour cells to the body. Studies have shown that the overexpression of RAB1A in cancer cells may promote autophagy progression by effecting with optic nerve protein (OPTN, an autophagy

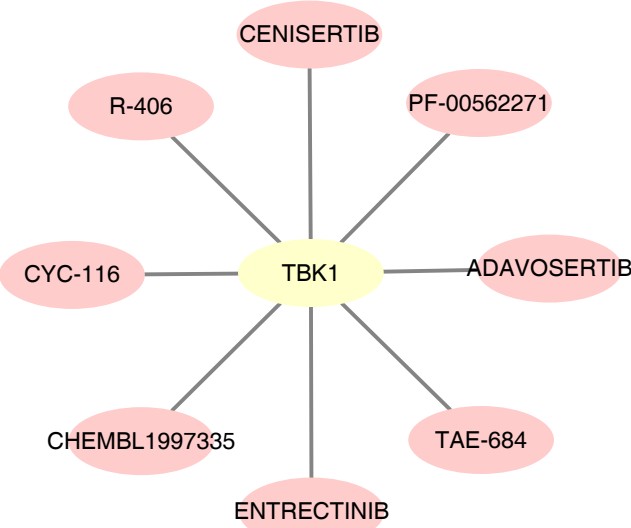

**Figure 9** **Potential drugs targeting feature genes.** Yellow represents gene and pink represents drug.

receptor). The knockout of RAB1A not only prevents endoplasmic reticulum-Golgi transport but also inhibits autophagy formation (*Song et al., 2018*). Meanwhile, RAB1A interacts with C9orf72 to regulate the initiation of autophagy by regulating the transport of the ULK1 autophagy initiation complex to phagocytic cells (*Webster et al., 2016*). A recent study reported that RAB1A plays a role in autophagy by recruiting and directly activating autophagy-specific VPS34 complex I (VPS34/VPS15/Beclin 1/ATG14L) (*Tremel et al., 2021*). However, most existing studies on RAB1A are regarding its role in tumours, with hardly any studies on its role in NAFLD. In this study, the samples in NAFLD groups had a up-regulated levels of RAB1A, which might be used as a reference for future studies on its role in NAFLD.

GOPC plays a role in intracellular protein trafficking and degradation. It also regulates the intracellular trafficking of the ADR1B receptor and plays a role in autophagy (*He et al., 2004*; *Cheng et al., 2002*). In this study, the abnormal up-regulation of GOPC expression was observed in NAFLD, while, very few studies on GOPC exist, thus, we need more experiments to explore its function in NAFLD. In our study, the TBK1, RABA1,GOPC were confirmed higher expressed in NAFLD through rats models and *in silico*. Previous studies have confirmed the importance of TBK1 in the development of NAFLD. Some related pharmacological treatments refer to TBK1 in NAFLD have also been reported. Although there are no studies research about RABA1 and GOPC in NAFLD, but we still thought RABA1 and GOPC may be plays important role in the occurrence of NAFLD. This study provides some theoretical basis for in-depth research between RABA1 and GOCP with NAFLD.

Currently, the pathogenesis of NAFLD is yet to be wholly elucidated. The pathogenesis of NAFLD involves a variety of factors, such as environmental factors, obesity, changes

in microbiota and susceptibility to gene mutations, among which imbalance of pro-inflammatory and anti-inflammatory cytokines within adipose tissue plays a key role in the pathological process of NAFLD (*Arab, Arrese & Trauner, 2018*; *Paredes-Turrubiarte et al., 2016*). In the study, immune infiltration analysis demonstrated that the different proportions of astrocytes, naïve CD4+ T cells, chondrocytes, epithelial cells, iDC, endothelial cells, macrophages and smooth muscle might be involved in NAFLD's pathogenesis. Macrophage is an important component of liver inflammation, with activated macrophages secreting various pro-inflammatory cytokines, such as TNF- $\alpha$, TGF- $\beta$1 and IL-6 (*Paredes-Turrubiarte et al., 2016*; *Kakino et al., 2018*). Notably, a recent study demonstrated that macrophages can induce hepatic TBK1 activation and expression (*Zheng et al., 2021*), which is consistent with our findings. Our study revealed that TBK-1, RAB-1A and GOPC were positively correlated with macrophages. Additionally, M2 Kupffer cells can promote the apoptosis of M1 Kupffer cells by secreting IL-10, consequently inhibiting the development of NASH (*Wan et al., 2014*). Dendritic cells can reduce the inflammatory reaction of NASH by clearing necrotic fragments and apoptotic bodies in the liver (*Henning et al., 2013*). The current study demonstrated a negative correlation between iDCs and the expression of TBK-1, RAB-1A and GOPC. The number of NKT cells is reduced in the presence of moderate to severe fatty inflammation, and the upregulation of NKT cells in the liver can relieve the hepatic steatosis induced by a high-fat diet. When NKT cells are over-depleted, the metabolic changes subsequently lead to NASH progression (*Tang et al., 2022*). In our study, TBK1, RAB1A and GOPC also showed a significant positive correlation with the expression of astrocytes and smooth muscle. The study of a single inflammatory factor and its interaction mechanism in NAFLD will aid in elucidating its pathogenesis in NAFLD. Currently, pro-inflammatory factor-related inhibitors have entered the clinical trial stage and are expected to become specific drugs for the treatment of NAFLD.

TFs are specific sites in the promoter sequence of target genes that regulate the expression level of downstream genes on a pre-transcriptional level by binding to the specific sequence motif (*Zacksenhaus et al., 2017*; *Zhang, Najmi & Schneider, 2017*). A large number of related studies have reported that TFs play an important role in the transcriptional regulatory network. Approximately 32 TFs can regulate ARGs to control autophagy in NAFLD (*Ueno & Komatsu, 2017*; *Di Malta, Cinque & Settembre, 2019*). Notably, cAMP response element-binding protein (CREB) and forkhead box O proteins are the most representative TFs and can upregulate autophagy genes and influence glucose and lipid metabolism (*Seok et al., 2014*; *Xiong et al., 2012*). In our study, 124 TF-mRNAs pairs of feature genes were identified, among which CREB-1 was associated with the three feature genes. Notably, CREB-1 has been reported to alleviate NAFLD *via* the CREB pathway (*Xu et al., 2022*).

The therapeutic agents which target TFs could prevent the progression of NAFLD into NASH. This study predicted eight therapeutic agents, which are based on TBK1, for the treatment of NAFLD. R-406 is a spleen tyrosine kinase inhibitor that can inhibit the SKY signalling pathway induced by LPS- and IFN- $\gamma$ in macrophages. R-406 is also upregulated in hepatocytes (*Qu et al., 2018*). Moreover, it reduces immune complex-mediated inflammation through dose-dependently inhibited nitric-oxide release and

M1-specific markers in M1-differentiated macrophage. It can also significantly relieve liver inflammation, which could be a potential therapeutic approach for the treatment of NASH (*Kurniawan et al., 2018*). However, studies on the other seven therapeutic agents identified herein are scarce, remains to be further examined.

This study has certain shortcomings. Firstly, the sample size of the training set is small, which requiring subsequent larger cohorts to verify the expression and diagnostic performance of three feature genes. Secondly, further functional experiments *in vivo* and *in vitro* targeting the feature genes are required for exploring the causality as well as exact effect of the identified genes on the pathogenesis of NAFLD. Thirdly, the connections as well as underlying regulatory mechanism of predicted TFs binding sites and m6A-related autophagy genes are necessary to examine through more studies.

## CONCLUSIONS

In conclusion, three m6A-related autophagy genes, namely TBK1, RAB1A and GOPC, were considered to be relevant to NAFLD progression based on various bioinformatic analyses. Differences of the expression levels of key genes between case and control samples were examined through two online NAFLD-related cohorts and animal models. The strong correlations of three key with smooth muscle, endothelial cells were explored as well. The potential TF binding sites and drugs targeting three genes were predicted, which initially provides a systematically comprehensive analysis of m6A-related autophagy genes in NAFLD and predicts potential agents for the treatment of NAFLD. Nonetheless, further exploration of m6A-related autophagy genes would aid in elucidating the mechanism of occurrence in NAFLD, thereby improving NAFLD treatment options.

## ACKNOWLEDGEMENTS

We would like to thank all the reviewers who participated in the review and BULLET Editor for its linguistic assistance during the preparation of this manuscript.

### Funding
This work was supported by the National Natural Science Foundation of China (grant number 81960440 and grant number 82070594). The funders had no role in study design, data collection and analysis, decision to publish, or preparation of the manuscript.

### Grant Disclosures
The following grant information was disclosed by the authors:
National Natural Science Foundation of China: 81960440, 82070594.

### Competing Interests
The authors declare there are no competing interests.
## Author Contributions

- Ziqing Huang conceived and designed the experiments, analyzed the data, prepared figures and/or tables, and approved the final draft.
- Linfei Luo performed the experiments, prepared figures and/or tables, and approved the final draft.
- Zhengqiang Wu performed the experiments, prepared figures and/or tables, and approved the final draft.
- Zhihua Xiao analyzed the data, authored or reviewed drafts of the article, and approved the final draft.
- Zhili Wen conceived and designed the experiments, analyzed the data, authored or reviewed drafts of the article, and approved the final draft.

## Animal Ethics

The following information was supplied relating to ethical approvals (*i.e.*, approving body and any reference numbers):

The animal experiment was approved by the ethical review committee of The Second Affiliated Hospital of Nanchang University (approval number: NCULAE-20221031008).

## Data Availability

The data is available at GEO: GSE66676 and GSE130970. The PCR data are available in the Supplemental Files.

## Supplemental Information

Supplemental information for this article can be found online at http://dx.doi.org/10.7717/peerj.17011#supplemental-information.

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
