# Peer review of "Identification of m6A-associated autophagy genes in non-alcoholic fatty liver"

_PeerJ, doi:10.7717/peerj.17011_

## Round 0.1 · original submission · Major Revisions

Please address concerns of the reviewers and amend manuscript accordingly. In particular, R2 has serious concerns which have prevented them for adequately assessing your work, as well as stating that experimental verification is needed

Reviewer 1 ·

Basic reporting

1. Some for the language and type error need to be improved, for example:
1) Line 28: “ly endothelial cells”;
2) Line 48-49: “However, the dangers and severity of NAFLD are underappreciated.”
3) Line 122 & Line 149: “HFD” & “WB” should be full spelled.
4)
2. Intro & Background:
1) Suggest to introduce mechanisms/strategy of current drug development for NAFLD and lead to the unmet medical needs for explore new mechanisms/targets.
2) Line 70-76: this is not introduction & Background but the summary of the study. Suggest to move this part.
3. Clear description notes should be added for all the A, B, C….. of all the figures, otherwise it’s very difficult to be understood.

Experimental design

Experimental Design is fine.

Validity of the findings

1. The manuscript report a novel result providing a reference for further research of NAFLD
2. Underlying data have been provided but not clear enough, for example:
a) Line 185-187, please explain clearly how to conclude the genes were obtained through the correlation analysis.
b) Line 188-190: please clarify what Figure 3C stand for?
c) Line 193-196: it’s unclear description of the result related Figure 3E.
d) Line 205-207, what’s the difference between the 2 figures of Figure 4D, representing different datasets? Please descript clearly.
e) In 3.8 Drug sensitivity analysis part, please also descript the results based on GOPC and RAB1A, even no predicted therapeutic agent could be identified.
3. Conclusions are well stated, linked to original research question & limited to supporting results.

Additional comments

NA

Reviewer 2 ·

Basic reporting

Writing:
Some sentences are written in rather awkward English. I suggest consulting native speakers to help revise the language, mostly the wording, for the revision.
Figures:
Most of figure panels are hard to understand.
Most of the figure labels are illegible.
No figure legends were provided.

Figure-by-figure comments:
Fig. 1
The heatmaps are not properly labeled. It is hard to understand what each row and column means.
Fig. 2
It is completely unclear to me how the co-expression network is generated from the pair-wise correlation analyses. Can the author clarify this procedure?
How are the raw results for panel C obtained? Bioinformatically or experimentally?
Fig. 5
Number of replicates for the western blotting were not reported.

Experimental design

Details of most of the bioinformatics analyses as well as packages were not provided, prohibiting the reviewer from properly judging the validity of the study.

Validity of the findings

Major weakness of the study is lack of true experimental validation of the causality of the identified genes with the pathogenesis of the disease. As a reuslt, most of the conclusions stays as correlative.

Additional comments

Non-alcoholic fatty liver disease (NAFLD) is a prevalent cause of chronci liver disease and is considered a great public health challenge. Understanding the mechanisms underlying the pathogenesis of such disease would greatly benefit global health. This study hypothesized that post-transcriptional regulation of autophagy-related genes, in particular m6A-modification of mRNAs plays essential roles in the pathogenesis of NAFLD. Through bioinformatic analysis and experimental validation, the study identified m6A-related autophagy genes that are differentially expressed in NAFLD samples, and confirmed the role of several key genes using through animal models.
Overall, this study raised a potentially important problem, but the study per se is poorly designed and executed.

---

## Round 0.2 · Minor Revisions

Please address the remaining concerns of reviewer #2 and amend the manuscript accordingly.

Reviewer 1 ·

Basic reporting

no additional comments

Experimental design

no additional comments

Validity of the findings

no additional comments

Additional comments

no additional comments

Reviewer 2 ·

Basic reporting

The author has properly addressed most of my previous comments. I have no further comments.

Experimental design

The author has properly addressed most of my previous comments. I have no further comments.

Validity of the findings

The author has toned down their conclusion by stating clearly the limitations of the study. It would be nice If the author can emphasize the correlative nature of the study in the abstract.

---

## Round 0.3 · accepted · Accept

All remaining concerns of the reviewers were adequately addressed and revised manuscript is acceptable now.